mathematical finance/complexity/mathematical modelling

agent-based learning, volatility smiles, trading, Black–Scholes and social learning

**Author for correspondence:**
Tushar Vaidya
e-mail: tushar_vaidya@sutd.edu.sg

# Learning agents in Black–Scholes financial markets

Tushar Vaidya[1], Carlos Murguia[2] and Georgios Piliouras[1]

[1]Singapore University of Technology and Design, 8 Somapah Road, Singapore 487372, Singapore
[2]Eindhoven University of Technology, Department of Mechanical Engineering, 5612 AZ Eindhoven, Netherlands

TV, 0000-0002-2264-2595; CM, 0000-0002-8887-1390; GP, 0000-0002-6236-3566

Black–Scholes (BS) is a remarkable quotation model for European option pricing in financial markets. Option prices are calculated using an analytical formula whose main inputs are strike (at which price to exercise) and volatility. The BS framework assumes that volatility remains constant across all strikes; however, in practice, it varies. How do traders come to learn these parameters? We introduce natural agent-based models, in which traders update their beliefs about the true implied volatility based on the opinions of other agents. We prove exponentially fast convergence of these opinion dynamics, using techniques from control theory and leader-follower models, thus providing a resolution between theory and market practices. We allow for two different models, one with feedback and one with an unknown leader.

## 1. Introduction

Econophysics divides into two paradigms. Statistical Econophysics relies on data, fitting certain power laws to existing asset prices at various time scales [1,2]. In statistical Econophysics, zero-intelligence agents have random interactions. Agents are homogeneous and have no learning ability. The central object of study is historical price data. The viewpoint is that interacting zero-intelligence traders' actions are already incorporated into price fluctuations. The focus is on the macroscopic aggregation of interactions in the form of available data.

While this is an important area of research, agent-based Econophysics offers the opportunity to study the microscopic interactions in more detail, where agents are heterogeneous.

Our objective is to offer a cogent and clear motivation for agent-based Econophysics in the context of option volatilities, whereby learning and interaction are made explicit. To an outsider, it may seem that financial assets are observed at one price, decided by the market. In reality, prices fluctuate throughout the day and there is no equilibrium price: it is

always in flux. Interaction between strategic traders and other players is embedded in all transactions and informational channels. Interaction is vital to understanding markets. The motivation for this paper was inspired by the works of Kirman [3] and Follmer *et al.* [4]. Rather than develop a thorough game theoretic or mean-field model, we advocate something in between. We aim to take a more nuanced view of agent-based Econophysics as espoused by Chakraborti *et al.* [5].

## 1.1. Our contribution

We introduce two different classes of learning models that converge to a consensus. Our interest is not in equilibrium but what process leads to it [6–8]. The first introduces a feedback mechanism (§4.1, theorem 4.1) where agents who are off the true 'hidden' volatility parameter feel a slight (even infinitesimally so) pull towards it along with the all the other 'random' chatter of the market. This model captures the setting where traders have access to an alternative trading venue or an information source provided by brokers and private message boards. The second model incorporates a market leader (e.g. Goldman Sachs) that is confident in its own internal metrics or is privy to client flow (private information) and does not give any weight to outside opinions (§4.3, theorem 4.4). Proving the convergence results (as well as establishing the exponentially fast convergence rates) requires tools from discrete dynamical systems. We showcase as well as complement our theoretical results with experiments (e.g. figure 2*a–d*), which for example show that if we move away from our models, convergence is no longer guaranteed.

We formalize the multi-dimensional analogues of our two models by using Kronecker products (§5, theorems 5.1 and 5.3). Thus, our models show how a volatility curve could function as a global attractor given adaptive agents. We conclude the paper by discussing future work and connections to other fields.

# 2. Derivatives and social learning

Before discussing the main models of this paper, we give an overview of options markets and trading. We then motivate our framework and explain why certain social learning models are appropriate.

## 2.1. Trading

Most trading is done electronically. To be dominant, firms now invest huge sums in technology to get an edge. For futures trading, speed is vital to profits. Trading complex derivatives requires not only speed but huge amounts of investment in quantitative models. This, in turn, feeds the need for mathematicians, computer scientists and engineers. Increasingly, over the last two decades, the way trading is conducted has also seen drastic changes. Electronification of the markets has affected both instruments traded on and off exchange. Algorithmic trading drives not only plain vanilla instruments like stocks and futures but also derivatives [9–11]. Furthermore, the distinction between stock exchanges and over-the-counter (OTC) markets is not as clear as it once was [12]. In OTC markets, trading is between two counterparties and there is no centralized marketplace. Increasingly, over the last decade, there has been a regulatory push to make OTC markets more exchange-like. In OTC markets, participants may see what their competitors are quoting for a particular security, but volume and the actual price transacted remain the privy of the bilateral counterparties. In some quarters, OTC markets are usually referred to as being quote-driven or truly dark markets [13]. Regulation in the USA and European Union has resulted in fragmented exchange-based trading but centralization of opaque OTC markets.

## 2.2. Options markets

Derivative contracts are actively traded across the world's financial markets with a total estimate value in the trillions of dollars. To get an intuitive understanding of the setting and the issues at hand, let us consider the prototypical example of European options.

A European option is the right to buy or sell an underlying asset at some point in the future at a fixed price, also known as the strike. A call option gives the right to buy an asset and a put option gives the right to sell an asset at the agreed price. On the opposite side of the buyer is the seller who has relinquished his control of exercise. Buyers of puts and calls can exercise the right to buy or sell. Sellers of options have to fulfil obligations when exercised against. The payoff of a buyer of a call

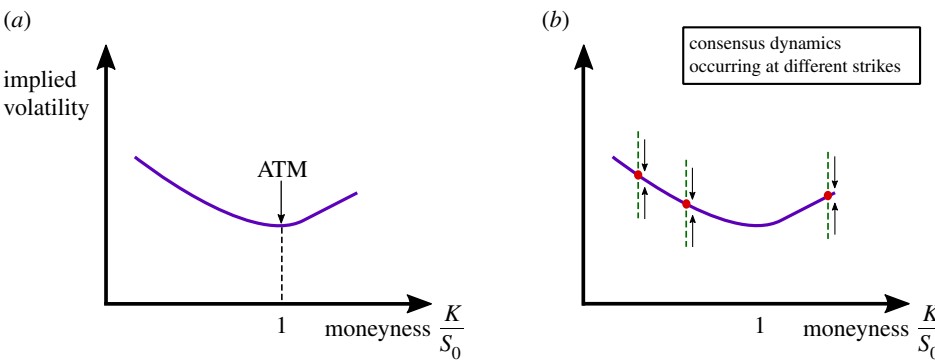

**Figure 1.** (*a*) A typical implied volatility smile for varying strikes *K* divided by fixed spot price. Moneyness is $K/S_0$. ATM denotes At-The-Money where *K* equals $S_0$. (*b*) Consensus occurs as all traders' opinions of the implied volatility converge, round by round, to a distinct value for varying strikes.

option with stock price $S_T$ at expiry time *T* and exercise price *K* is max$\{S_T - K, 0\}$, whereas for a put option is max$\{K - S_T, 0\}$.

To get a price, we input the current stock price $S_0$ (e.g. \$101), the exercise price *K* (e.g. \$90), the expiry *T* (e.g. three months from today) and the volatility $\sigma$ in the Black–Scholes (BS) formula [14–16]:

$$\text{price} = \text{BS}(S_0, K, T, \sigma).$$

Volatility, which captures the beliefs about how turbulent the stock price will be, is left up to the market. This parameter is so important that in practice the market trades European calls and puts by quoting volatilities.[1]

Options can be struck at different strike prices on the same asset (e.g. *K* = \$90, \$75, \$60). If the underlying asset and the time to exercise *T* (e.g. three months) are the same, one would expect the volatility to be the same at different strikes. In practice, however, the market after the 1987 crash has evolved to exhibit different volatilities. This rather strange phenomenon is referred to as the smile, or smirk (figure 1). Depending on the market, these smirks can be more or less pronounced. For instance, equity markets display a strong skew or smirk. A symmetric smile is more common in foreign exchange options markets. An excellent introduction to volatility smiles is given in [17].

How does the market decide what the quoted volatility should be (e.g. for a stock index three months from now)? This is a critical but not well-understood question. This is exactly what we aim to study by introducing models of learning agents who update their beliefs about the volatility. Agent-based models on volatility–smile interaction and formation have not been thoroughly addressed in finance or Econophysics. They remain a challenge [18]. Previous attempts have been made, but the focus has never been on the mathematical or specific nature of interaction [19,20]. Furthermore, our work takes into account the physicality of how trading occurs. An alternative perspective is offered in [21,22], again though the nature of interaction is missing. Nevertheless, these early attempts offer a good indication that at least the problem has garnered significant interest in different disciplines.

## 2.3. Econophysics

The challenge for physicists is not to force existing physics-based models on human behaviour but rather develop new models [23–25]. To go from local microscopic interactions to global macroscopic behaviour is not an easy task [26,27]. In fact, the choice of models seems infinite. There are a plethora of agent-based models [5,25,28]. Which one is correct? And, moreover, which type of social learning is representative of financial markets trading? LeBaron provides an early guide [29]. Agent-based models were proclaimed as the future for Econophysics [30,31]. While development in this area has been steady, the problem of the emergence of volatility smiles remains unresolved. The volatility smile is an active and vigorous area of research in the mathematical finance community [32–34]. Many models postulate a stochastic process for the underlying stock and volatility combined.

---

[1]Using the BS formula with a particular implied volatility, traders obtain a dollar value for the price.

## 2.4. Knightian uncertainty

Risk and uncertainty are two different concepts [35–37]. Risky assets are those on which the probabilities of random events are well defined and known. For instance, suppose we observe historical data of a stock price. Are we confident to claim we know the distribution of the stock's returns? If we are, then the stock is considered risky. Its risk is quantifiable. However, if we were unsure of even the correct probability measure, then we would be faced with uncertainty. In a sense, this captures the essence of financial markets. Traders and players use different probability measures when trading and quoting options. No single measure dominates. In fact, there are many models that are consistent with the observation of a finite number of strike volatilities in the market [38–41]. In practice, the choice of a correct probability measure such that a derivative contract is priced correctly is a subjective and quantitative exercise. In any case, no perfect model exists [42–46]. As a result, participants in financial markets are free to choose whichever probability model they calibrate to market data [47–49].

The problem with economics-based models and those in mathematical finance literature is that many times the analysis is centred on a representative agent. In the case of risk and uncertainty, the choice of pricing a derivative contract reduces to choosing a correct equivalent martingale measure under which a derivative claim is replicable. For market-makers and dealers, the choice of models is vast. Each player has to make a choice and inevitably no two institutions will use the same models with the same parameters. In this case, it is remarkable that the market will aggregate the diverse beliefs to arrive at a consensus smile. At the microscopic level, though, the dealers are observing one another's updates. Hence, our model can be seen as a meta-opinion dynamics framework built upon the individual choices of the dealers.

## 2.5. Non-Bayesian financial markets

In financial markets, updating occurs at high frequency across geographical locations [50,51]. Agents move simultaneously: cancellations are the norm [52–54]. In practical terms, sequential Bayesian learning models do not seem appropriate [55,56]. Bayesian observational learning examples include [57–59]. These models are *sequential* in nature. They study herd behaviour. As time passes, a player in turn observes the actions of previous agents and receives a private signal. Each agent has a one-off decision when she updates her posterior probability and takes an action. In some instances, the $n$th agent may reach the truth as $n \to \infty$.

In DeGroot learning, myopic updating occurs in each iteration. Agents in our set-up have fixed weights but update their responses until consensus is reached. Recently, there have been some experimental papers on the evidence of DeGroot updating [60,61]. Repeated averaging models are our base precisely because they capture the nature of interaction and learning in financial markets so compactly. Players can observe previous choices but not the payoffs of their competitors. A more in-depth discussion of learning in games would take us further away from our goal of studying the mathematical nature of interaction. The reader can consult [62,63] for a game-theoretic perspective.

# 3. Model description

In mathematical opinion dynamic models, agents take views of other agents into account before arriving at their own updated estimate. Agents can observe other agents' previous signals.

DeGroot [64] was one of the early developers of such observational learning dynamics. While simple, these models allow us to examine convergence to consensus. In a sense, these types of models are called naive models, as agents can recall perfectly what the other players submitted in the previous round. See the survey papers [65–68].

## 3.1. Volatility basics

Agents have an initial opinion of the implied volatility, which they update after taking into account volatilities of other agents. A feedback mechanism aids the agents in arriving at the true volatility parameter.

At all times, the focus is on a static picture of the volatility smile. Within this static framework agents are updating their opinion of the true implied volatility. This updating occurs in a high-frequency sense. In an exchange setting, one can think of all bids and offers as visible to agents. The agents initially are

unsure of the true value of the implied volatility, but by learning—and feedback—reach consensus on the true parameter. Our first attempt is a naive learning model common in social networks. Learning occurs between trading times. Therefore, our implicit assumption is that no transactions occur while traders are adjusting and learning each other's quotes.

This rather peculiar feature is market practice. Trading happens at longer intervals than quote updating. This is as true for high-frequency trading of stocks as it is for options markets. Quotes and prices—or rather vols—are changing more frequently than actual transactions.

Each dollar value of an option corresponds to an implied volatility parameter $\sigma(K, T) \in (0, 1)$ that depends on strike and expiry. Implied volatility is quoted in percentage terms.

**Assumption 3.1.** We have three types of players: agents/traders, brokers and leaders. Brokers give feedback to the traders. The ability of agents to determine this feedback is their learning ability. Leaders are unknown and do not give feedback but their quotes are visible.

## 3.2. Naive opinion dynamics

A first approach towards opinion dynamics is to assume each agent takes a weighted average of other agents' opinions and updates his own estimate of the volatility parameter for the next period. At time $t$, the opinion $x_t^i \in \mathbb{R}$ of the $i$-th agent is given by

$$x_t^i = \sum_{j=1}^{n} a_{ij} x_{t-1}^j, \quad t \in \mathbb{N}, \tag{3.1}$$

where $x_{t-1}^j \in \mathbb{R}$ is the opinion of agent $j$ at time $(t-1)$ and $a_{ij} \geq 0$ denotes the opinion weights for the $n$ players with $\sum_{j=1}^{n} a_{ij} = 1$ and $a_{ii} > 0$ for all $1 \leq i \leq n$. Define $X_t := (x_t^1, \ldots, x_t^n)^\top$, then the opinion dynamics of the $n$ agents can be written in matrix form as follows:

$$X_t = AX_{t-1}, \tag{3.2}$$

where $A := a_{ij} \in \mathbb{R}^{n \times n}$ is a *row-stochastic matrix*.

**Definition 3.2 (consensus).** The $n$ agents (3.2) are said to reach consensus if for any fixed initial condition $X_1 \in \mathbb{R}^n$, $|x_t^i - x_t^j| \to 0$ as $t \to \infty$ for all $i, j \in \{1, \ldots n\}$.

**Definition 3.3 (consensus to a point).** The $n$ agents (*3.2*) are said to reach consensus to a point if for any initial condition $X_1 \in \mathbb{R}^n$, $\lim_{t \to \infty} X_t = c\mathbf{1}_n$, where $\mathbf{1}_n$ denotes the $n \times 1$ vector composed of only ones and $c \in \mathbb{R}$. The constant $c$ is often referred to as the consensus value.

For the opinion dynamics (3.2), we introduce the following result by [64] (see also [69] for definitions).

**Proposition 3.4.** *Consider the opinion dynamics in equation* (3.2). *If $A$ is aperiodic and irreducible, then for any initial condition $X_1 \in \mathbb{R}^n$ consensus to a point is reached. The consensus value c depends on both the matrix A and the initial condition $X_1$.*

**Remark 3.5.** Proposition 3.4 implies that if the row stochastic opinion matrix $A$ is aperiodic and irreducible, then all the agents converge to some consensus value $c$. However, since $c$ depends on the unknown initial opinion $X_1$, the consensus value $c$ is unknown and, in general, different from the true volatility $\sigma(K, T)$. We wish to alleviate this and thus introduce two novel models.

# 4. Consensus (scalar agent dynamics)

In this section, we assume that the agents are able to learn how far off they are from the true volatility by informational channels in the marketplace. There are many avenues, platforms and private online chat rooms that provide quotes for option prices; some of these are stale and some are fresh. The agents' learning ability determines the quality of the feedback from all these sources. In reality, options are not traded on one exchange or platform. There are multiple venues and, though there might be a dominant marketplace, the same instruments can be traded across different venues and locations. We aggregate all of this information in the form of feedback with learning ability. If agents are fast learners, they adjust their volatility estimates quickly.

## 4.1. Consensus with feedback

We model this feedback by introducing an extra driving term into the opinion dynamics (3.1). An early model developed by Mizuno *et al.* [70] shares some similarities to ours. Traders use feedback from past behaviour. Our model is a discrete autoregressive process but the focus is on learning in high-frequency time [71]. Furthermore, our model formalizes this in a more social and dynamical set-up. In particular, we feed back the difference between the agents' opinion and the true volatility $\sigma(K, T)$ scaled by a *learning coefficient* $\epsilon_i \in (0, 1)$. We assume that $\sigma(K, T)$ is invariant, i.e. for some fixed $\bar{\sigma} \in (0, 1)$, $\sigma(K, T) = \bar{\sigma}$ for some fixed strike $K$ and maturity $M$. Then the new model is written as follows:

$$x_t^i = \sum_{j=1}^n a_{ij} x_{t-1}^j + \epsilon_i(\bar{\sigma} - x_{t-1}^i), \tag{4.1}$$

or in matrix form

$$X_t = A X_{t-1} + \mathcal{E}(\bar{\sigma}\mathbf{1}_n - X_{t-1}), \tag{4.2}$$

where $\mathcal{E} := \mathrm{diag}(\epsilon_1, \ldots, \epsilon_n)$. Then we have the following result.

**Theorem 4.1.** *Consider the agent dynamics in (4.2) and assume that $\epsilon_i \in (0, a_{ii})$, $i = \{1, \ldots, n\}$. Then consensus to $\bar{\sigma}$ is reached, i.e. $\lim_{t \to \infty} X_t = \bar{\sigma}\mathbf{1}_n$.*

*Proof.* It is easy to verify that the solution $X_t$ of the difference equation (4.2) is given by

$$X_{t+1} = (A - \mathcal{E})^t X_1 + \sum_{j=0}^{t-1} (A - \mathcal{E})^j \mathcal{E}\bar{\sigma}\mathbf{1}_n, \quad t > 1. \tag{4.3}$$

By the Gershgorin circle theorem, the spectral radius $\rho(A - \mathcal{E}) < 1$ for all $i$, $\epsilon_i < a_{ii}$. It follows that $\sum_{j=0}^\infty (A - \mathcal{E})^j \mathcal{E}\bar{\sigma}\mathbf{1}_n = (I_n - A + \mathcal{E})^{-1}\mathcal{E}\bar{\sigma}\mathbf{1}_n$, where $I_n$ denotes the identity matrix of dimension $n$, and $\lim_{t \to \infty} (A - \mathcal{E})^t = \mathbf{0}$, see [72]. As the matrix $A$ is row stochastic, $(I - A)\mathbf{1}_n = \mathbf{0}_n$, where $\mathbf{0}_n$ denotes the $n \times 1$ vector composed of only zeros. Hence, we can write $\mathcal{E}\mathbf{1}_n = (I_n - A)\mathbf{1}_n + \mathcal{E}\mathbf{1}_n$, and consequently $\mathbf{1}_n = (I_n - A + \mathcal{E})^{-1}\mathcal{E}\mathbf{1}_n$. It follows that

$$\lim_{t \to \infty} X_{t+1} = \lim_{t \to \infty} (A - \mathcal{E})^t X_1 + \sum_{j=0}^\infty (A - \mathcal{E})^j \mathcal{E}\bar{\sigma}\mathbf{1}_n$$

$$= \mathbf{0}_n + (I_n - A + \mathcal{E})^{-1}\mathcal{E}\mathbf{1}_n\bar{\sigma} = \mathbf{1}_n\bar{\sigma},$$

and the assertion follows. ∎

**Corollary 4.2.** *Consensus to $\bar{\sigma}$ is reached exponentially with convergence rate $\|A - \mathcal{E}\|_\infty$, i.e. $\max_i \{\|x_t^i - \bar{\sigma}\|\} \leq \|A - \mathcal{E}\|_\infty^{t-1}\|X_1 - \bar{\sigma}\mathbf{1}_n\|_\infty$, $i \in \{1, \ldots, n\}$, where $\|\cdot\|_\infty$ denotes the matrix norm induced by the vector infinity norm.*

*Proof.* Define the error sequence $E_{t-1} := (X_{t-1} - \bar{\sigma}\mathbf{1}_n) \in \mathbb{R}^n$. Then, from (4.2), the following is satisfied:

$$E_t = X_t - \bar{\sigma}\mathbf{1}_n$$
$$= A X_{t-1} + \mathcal{E}(\bar{\sigma}\mathbf{1}_n - X_{t-1}) - \bar{\sigma}\mathbf{1}_n$$
$$= A(E_{t-1} + \bar{\sigma}\mathbf{1}_n) + \mathcal{E}(\bar{\sigma}\mathbf{1}_n - (E_{t-1} + \bar{\sigma}\mathbf{1}_n)) - \bar{\sigma}\mathbf{1}_n$$
$$= (A - \mathcal{E})E_{t-1} + \bar{\sigma}(A - I_n)\mathbf{1}_n$$
$$= (A - \mathcal{E})E_{t-1}.$$

The last equality in the above expression follows from the fact that $(A - I_n)\mathbf{1}_n = 0$, because $A$ is a stochastic matrix. The solution $E_t$ of the above difference equation is given by $E_t = (A - \mathcal{E})^{t-1}E_1$, where $E_1 = X_1 - \bar{\sigma}\mathbf{1}_n$ denotes the initial error. Let $\|E_t\|_\infty = \max_i (\|e_t^i\|)$, $i \in \{1, \ldots, n\}$, where $E_t = (e_t^1, \ldots, e_t^n)^{\mathsf{T}}$. Note that exponential convergence of $\|E_t\|_\infty$ implies exponential convergence of $E_t$ itself. With the solution $E_t = (A - \mathcal{E})^{t-1}E_1$, the following can be written:

$$\|E_t\|_\infty = \|(A - \mathcal{E})^{t-1}E_1\|_\infty$$

$$\leq \|(A - \mathcal{E})\|_\infty^{t-1}\|E_1\|_\infty,$$

where $\|(A - \mathcal{E})\|_\infty$ denotes the matrix norm of $(A - \mathcal{E})$ induced by the vector infinity norm [72]. The inequality $\|E_t\|_\infty \leq \|(A - \mathcal{E})\|_\infty^{t-1}\|E_1\|_\infty$ implies exponential convergence if $\|(A - \mathcal{E})\|_\infty < 1$. Because $A = a_{ij}$ and $\mathcal{E} = \mathrm{diag}(\epsilon_1, \ldots, \epsilon_n)$, we can compute $\|(A - \mathcal{E})\|_\infty$ as

$\|(A - \mathcal{E})\|_\infty = \max_i (\sum_{j=1, j \neq i}^n \|a_{ij}\| + \|a_i - \epsilon_i\|)$, $i \in \{1, \ldots, n\}$. The matrix $A$ is stochastic, which implies $a_{ij} \geq 0$ and $\sum_{i=1}^n \|a_{ij}\| = 1$. Therefore, under the conditions of theorem 4.1 (i.e. $\epsilon_i \in (0, \ a_{ii})$), $\|(A - \mathcal{E})\|_\infty = \max_i (\sum_{j=1, j \neq i}^n \|a_{ij}\| + \|a_i - \epsilon_i\|) < 1$ and hence exponential convergence of the consensus error $E_t$ can be deduced with rate given by $\|(A - \mathcal{E})\|_\infty = \max_i (\sum_{j=1, j \neq i}^n \|a_{ij}\| + \|a_i - \epsilon_i\|)$. ∎

## 4.2. Random case

Under suitable random conditions for the trust matrix $A$ and $\mathcal{E}$, we can still have consensus. In this case, the learning rates and weights are independently and identically distributed from each iteration. However, we need a condition to ensure convergence, namely that on average the learning rates are less than the self-belief condition. Since this is only in expectation, a probabilistic statement, there is some leeway on the learning rates being strictly less than self-belief $a_{ii}$ at time $t$.

**Theorem 4.3.** *Consider the updating rule*

$$X_t = A_t X_{t-1} + \mathcal{E}_t(\bar{\sigma} \mathbf{1}_n - X_{t-1}), \tag{4.4}$$

*where $A_t$ and $\mathcal{E}_t$ are independent and identically distributed (iid). Furthermore, suppose*

$$-\infty < \mathbb{E}[\log \|A_t - \mathcal{E}_t\|_\infty] < 0 \text{ and } \|X_0 - \bar{\sigma}\| < \infty,$$

*then consensus to $\bar{\sigma}$ is reached, i.e. $\lim_{t \to \infty} X_t = \bar{\sigma} \mathbf{1}_n$.*

*Proof.* We rewrite the above iteration, subtracting $\bar{\sigma}$ from both sides and dropping the one vector notation as the context is clear

$$
\begin{aligned}
X_t - \bar{\sigma} &= A_t X_{t-1} + \mathcal{E}_t(\bar{\sigma} - X_{t-1}) - \bar{\sigma}, \\
X_t - \bar{\sigma} &= A_t X_{t-1} - A_t \bar{\sigma} + \mathcal{E}_t \bar{\sigma} - \mathcal{E}_t X_{t-1}, \\
X_t - \bar{\sigma} &= (A_t - \mathcal{E}_t)(X_{t-1} - \bar{\sigma}), \\
Y_t &= (A_t - \mathcal{E}_t) Y_{t-1}
\end{aligned}
$$

and

$$Y_t = B_t Y_{t-1},$$

where $Y_t = X_t - \bar{\sigma}$ and $B_t = A_t - \mathcal{E}_t$. We want to show $Y_t \to 0$. To this end, iterating the above recursion gives us

$$Y_t = \underbrace{B_t B_{t-1} \cdots B_1}_{\text{iid matrices}} Y_0.$$

Taking norms on the above equation results in the following inequalities, understanding that we mean the $\|\cdot\|_\infty$ norm:

$$
\begin{aligned}
\|Y_t\| &= \|B_t B_{t-1} \cdots B_1 Y_0\|, \\
\|Y_t\| &\leq \|B_t\| \|B_{t-1}\| \cdots \|B_1\| \|Y_0\|, \\
\log \|Y_t\| &\leq \log(\|B_t\| \|B_{t-1}\| \cdots \|B_1\| \|Y_0\|), \\
\log \|Y_t\| &\leq \log \|B_t\| + \log \|B_{t-1}\| + \cdots + \log \|B_1\| + \log \|Y_0\|
\end{aligned}
$$

and

$$\|Y_t\| \leq \exp^{t \frac{\sum_{k=1}^t \log \|B_k\|}{t}} \|Y_0\|.$$

The first inequality follows by sub-multiplicative property of matrix norms. Moreover, by the law of large numbers $\frac{1}{t} \sum_{k=1}^t \log \|B_k\|_\infty \longrightarrow \mathbb{E}[\log \|A_t - \mathcal{E}_t\|_\infty]$, which is negative by assumption. So the exponent ensures that, as the initial opinion $\|Y_0\|_\infty < \infty$ is finite,

$$\lim_{t \to \infty} \|Y_t\|_\infty = 0.$$

Consequently, $Y_t \longrightarrow 0$ and every agent reaches consensus. ∎

Note we do not require the stronger condition that $\log \|A_t - \mathcal{E}_t\|_\infty < 0$, for all $t$. Unlike the deterministic case, the random case allows considerable flexibility. Neither self-belief $a_{ii} > 0$ nor positive learning $\epsilon_i$ is required for all times. However, there must be some interaction and learning for beliefs to converge. As matrix products do not commute, if we were to follow the full expansion of the recursion in any of the dynamics, the result would be long, unwieldy matrix products. Random matrix products and dynamics are an active area of research not only in mathematics but also in

physics and control theory [73–78]. While the random case is certainly interesting, in this article our focus is on the first steps of modelling interaction and learning dynamics.

## 4.3. Consensus with an unknown leader

One criticism of model (4.2) is that feedback, even if it is not perfect, has to be learned. In practice, there might not be a helpful mechanism that provides feedback. An alternative is to have an unknown leader embedded in the set of traders. The agents are unsure who the leader is but by taking averages of other traders, they all arrive at the opinion of the leader. In Markov chain theory, such behaviour is called an absorbing state. The leader guides the system to the true value. We assume that the *identity* of the leader is unknown to all agents.

Without loss of generality, we assume that the first agent (with corresponding opinion $x_t^1$) is the leader; it follows that $x_t^1 = \bar{\sigma}$, $a_{1i} = 0$, $i \in \{2, \dots, n\}$, and $a_{11} = 1$. Then in this configuration, the opinion dynamics is given by

$$X_t = AX_{t-1}, \quad A = \begin{pmatrix} 1 & 0 & \dots & 0 \\ a_{21} & a_{22} & \dots & a_{2n} \\ \vdots & \vdots & \dots & \vdots \\ a_{n1} & a_{n2} & \dots & a_{nn} \end{pmatrix} =: \begin{pmatrix} 1 & \mathbf{0} \\ * & \tilde{A} \end{pmatrix}, \quad (4.5)$$

with $a_{ij} \geq 0$, $\sum_{j=1}^{n} a_{ij} = 1$, $a_{ii} > 0$ for all $1 \leq i \leq n$, and for at least one $i \geq 2$, $\sum_{j=2}^{n} a_{ij} < 1$.

**Theorem 4.4.** *Consider the opinion dynamics in (4.5) and assume that the matrix $\tilde{A}$ is substochastic and irreducible. It holds that $\lim_{t \to \infty} X_t = \bar{\sigma}\mathbf{1}_n$, i.e. consensus to $\bar{\sigma}$ is reached.*

*Proof.* Define the invertible matrix $M \in \mathbb{R}^{n \times n}$

$$M := \begin{pmatrix} 1 & \mathbf{0} \\ \mathbf{1}_{n-1} & -I_{n-1} \end{pmatrix}.$$

Introduce the set of coordinates $\tilde{X}_{t-1} := MX_{t-1}$. Note that $\tilde{x}_{t-1}^1 = x_{t-1}^1$, $\tilde{x}_{t-1}^2 = x_{t-1}^1 - x_{t-1}^2, \dots, \tilde{x}_{t-1}^n = x_{t-1}^1 - x_{t-1}^n$. Hence, if the error vector $e_{t-1} := (\tilde{x}_{t-1}^2, \dots, \tilde{x}_{t-1}^n)^\top = \mathbf{0}_{n-1}$, then consensus to $x_t^1 = \bar{\sigma}$ is reached. Note that

$$MAM^{-1} = \begin{pmatrix} 1 & * \\ \mathbf{0} & \tilde{A} \end{pmatrix},$$

where $\mathbf{0}$ denotes the zero vector of appropriate dimensions and $\tilde{A}$ as defined in (4.5). By construction, $\tilde{X}_{t-1} := MX_{t-1} \to \tilde{X}_t = MX_t = MAX_{t-1} = MAM^{-1}\tilde{X}_{t-1}$; hence, the consensus error $e_t$ satisfies the following difference equation

$$\tilde{X}_t = MAM^{-1}\tilde{X}_{t-1} = \begin{pmatrix} 1 & * \\ \mathbf{0} & \tilde{A} \end{pmatrix}\tilde{X}_{t-1} e_t = \tilde{A}e_{t-1}, \quad (4.6)$$

and the solution of $e_t$ is then given by $e_t = \tilde{A}^t e_1$.

Because for at least one $i$, $\sum_{j=2}^{n} a_{ij} < 1$ and $\tilde{A}$ is substochastic and irreducible, the spectral radius $\rho(\tilde{A}) < 1$, see lemma 6.28 in [69]; it follows that $\lim_{t \to \infty} \tilde{A}^t = \mathbf{0}$. Therefore, $\lim_{t \to \infty} e_t = \mathbf{0}$ and the assertion follows. ∎

**Corollary 4.5.** *Let $\|\cdot\|_*$ denote some matrix norm such that $\|\tilde{A}\|_* < 1$ (such a norm always exists because $\rho(\tilde{A}) < 1$ under the conditions of theorem 4.4). Then consensus to $\bar{\sigma}$ is reached exponentially with the convergence rate given by $\|\tilde{A}\|_*$, i.e. $\max_i \{\|x_t^i - \bar{\sigma}\|\} \leq C\|\tilde{A}\|_*^{t-1}\|X_1 - \bar{\sigma}\mathbf{1}_n\|_\infty$, for $i \in \{1, \dots, n\}$ and some positive constant $C \in \mathbb{R}_{>0}$.*

*Proof.* See lemma 5.6.10 in [72] on how to construct such a $\|\cdot\|_*$. Now consider the consensus error $e_t$ defined in the proof of theorem 4.4, which evolves according to the difference equation (4.6). It follows that $e_t = \tilde{A}^{t-1}e_1$, where $e_1$ denotes the initial consensus error. Under the assumptions of theorem 4.4, $\rho(\tilde{A}) < 1$. By lemma 5.6.10 in [72], $\rho(\tilde{A}) < 1$ implies that there exists some matrix norm, say $\|\cdot\|_*$, such that $\|\tilde{A}\|_* < 1$. We restate the error with norms and obtain $\|e_t\|_\infty \leq \|\tilde{A}\|_*^{t-1}\|e_1\|_\infty$. Because all norms are equivalent in finite dimensional vector spaces (see ch. 5 in [72]), $\|e_t\|_\infty \leq \|\tilde{A}\|_\infty^{t-1}\|e_1\|_\infty \Rightarrow \|e_t\|_\infty \leq C\|\tilde{A}\|_*^{t-1}\|e_1\|_\infty$ for some positive constant $C \in \mathbb{R}_{>0}$. As $\|\tilde{A}\|_* < 1$, the norm of the consensus error $\|e_t\|_\infty$ converges to zero exponentially with rate $\|\tilde{A}\|_*$. ∎

# 5. Consensus (vectored agent dynamics)

In this section, we suppose that agents have beliefs over a range of strikes. Thus, each agent's opinion of the volatility curve is a vector with each entry corresponding to a particular strike. Typically, in markets, options are quoted for At-The-Money (ATM) $K = S_0$ and for two further strikes left of and right of the ATM level. Here, we examine the case of $k$ strikes and $n$ agents, i.e. each agent $i$ now has $k$ quotes for $k$ different moneyness levels. In this configuration, the true volatility is $\bar{\sigma} := [\sigma_1, \ldots, \sigma_k]^\top \in \mathbb{R}^k$. See figure 1b.

## 5.1. Consensus with feedback

Again, we assume that each agent takes a weighted average of other agents' opinions and updates his volatility estimate *vector* for the next period. At time $t$, the opinion $x_t^i \in \mathbb{R}^k$ of the $i$-th agent is given by

$$x_t^i = \sum_{j=1}^n a_{ij} x_{t-1}^j + \epsilon_i(\bar{\sigma} - x_{t-1}^i), \quad t \in \mathbb{N}, \tag{5.1}$$

where $\epsilon_i \in (0, 1)$ denotes the *learning coefficient* of agent $i$, $x_{t-1}^j \in \mathbb{R}^k$ is the opinion of agent $j$ at time $(t-1)$, and $a_{ij} \geq 0$ denotes the opinion weights for the $n$ agents with $\sum_{j=1}^n a_{ij} = 1$ and $a_{ii} > 0$ for all $1 \leq i \leq n$. In this case, the stacked vector of opinions is $X_t := (x_t^1, \ldots, x_t^n)^\top$, $X_t \in \mathbb{R}^{kn}$. The opinion dynamics of the $n$ agents can then be written in matrix form as follows:

$$X_t = (A \otimes I_k)X_{t-1} + (\mathcal{E} \otimes I_k)(\mathbf{1}_n \otimes \bar{\sigma} - X_{t-1}), \tag{5.2}$$

where $A = a_{ij} \in \mathbb{R}^{n \times n}$ is a *row-stochastic matrix*, $\mathcal{E} = \text{diag}(\epsilon_1, \ldots, \epsilon_n)$, and $\otimes$ denotes a Kronecker product. We have the following result.

**Theorem 5.1.** *Consider the opinion dynamics in (5.2) and assume that $\epsilon_i \in (0, a_{ii})$, $i = \{1, \ldots, n\}$. Then consensus to $\mathbf{1}_n \otimes \bar{\sigma}$ (with $\bar{\sigma} = [\sigma_1, \ldots, \sigma_k]^\top \in \mathbb{R}^k$) is reached, i.e. $\lim_{t \to \infty} X_t = \mathbf{1}_n \otimes \bar{\sigma}$.*

*Proof.* Define the error sequence $e_{t-1} := X_{t-1} - (\mathbf{1}_n \otimes \bar{\sigma})$. Note that $e_{t-1} = \mathbf{0}$ implies that consensus to $(\mathbf{1}_n \otimes \bar{\sigma})$ is reached. Given the opinion dynamics in (5.2), the evolution of the error $e_{t-1}$ satisfies the following difference equation:

$$\begin{aligned} e_t &= ((A - \mathcal{E}) \otimes I_k)X_{t-1} + ((\mathcal{E} \otimes I_k) - I_{kn})(\mathbf{1}_n \otimes \bar{\sigma}) \\ &= ((A - \mathcal{E}) \otimes I_k)e_{t-1} - (\mathbf{1}_n \otimes \bar{\sigma}) + (A \otimes I_k)(\mathbf{1}_n \otimes \bar{\sigma}) \\ &= ((A - \mathcal{E}) \otimes I_k)e_{t-1} + ((A - I_n)\mathbf{1}_n \otimes \bar{\sigma}). \end{aligned}$$

It is easy to verify that, because $A$ is stochastic, $(A - I_n)\mathbf{1}_n = \mathbf{0}_n$. Then the error dynamics simplifies to

$$e_t = ((A - \mathcal{E}) \otimes I_k)e_{t-1}, \tag{5.3}$$

and consequently, the solution $e_t$ of (5.3) is given by $e_t = ((A - \mathcal{E}) \otimes I_k)^t e_1$. By properties of the Kronecker product and Gershgorin's circle theorem, the spectral radius $\rho(A - \mathcal{E}) < 1$ for $\epsilon_i \in (0, a_{ii})$. It follows that $\lim_{t \to \infty} ((A - \mathcal{E}) \otimes I_k)^t = \mathbf{0}$, see [72]. Therefore, $\lim_{t \to \infty} e_t = \mathbf{0}_{kn}$ and the assertion follows. ∎

**Corollary 5.2.** *Consensus to $\bar{\sigma}$ is reached exponentially with the convergence rate given by $\|(A - \mathcal{E}) \otimes I_k)\|_\infty$, i.e. $\|X_t - (\mathbf{1}_n \otimes \bar{\sigma})\|_\infty \leq \|(A - \mathcal{E}) \otimes I_k)\|_\infty^{t-1} \|X_1 - (\mathbf{1}_n \otimes \bar{\sigma})\|_\infty$.*

The proof of the above result is very similar to previous corollaries and is omitted.

## 5.2. Consensus with an unknown leader

As in the scalar case, there is a leader driving all the other agents through the opinion matrix $A$. Again, without loss of generality, we assume that the first agent (with corresponding opinion $x_t^1 \in \mathbb{R}^k$) is the leader, $x_t^1 = \bar{\sigma} = [\sigma_1, \ldots, \sigma_k]^\top \in \mathbb{R}^k$, $a_{1i} = 0$, $i \in \{2, \ldots, n\}$, and $a_{11} = 1$. Then in this configuration, the opinion dynamics is given by

$$X_t = (A \otimes I_k)X_{t-1}, \quad A = \begin{pmatrix} 1 & 0 & \ldots & 0 \\ a_{21} & a_{22} & \ldots & a_{2n} \\ \vdots & \vdots & \ldots & \vdots \\ a_{n1} & a_{n2} & \ldots & a_{nn} \end{pmatrix} =: \begin{pmatrix} 1 & \mathbf{0} \\ * & \tilde{A} \end{pmatrix}, \tag{5.4}$$

with $a_{ij} \geq 0$, $\sum_{j=1}^n a_{ij} = 1$, $a_{ii} > 0$ for all $1 \leq i \leq n$, and for at least one $i \geq 2$, $\sum_{j=2}^n a_{ij} < 1$.

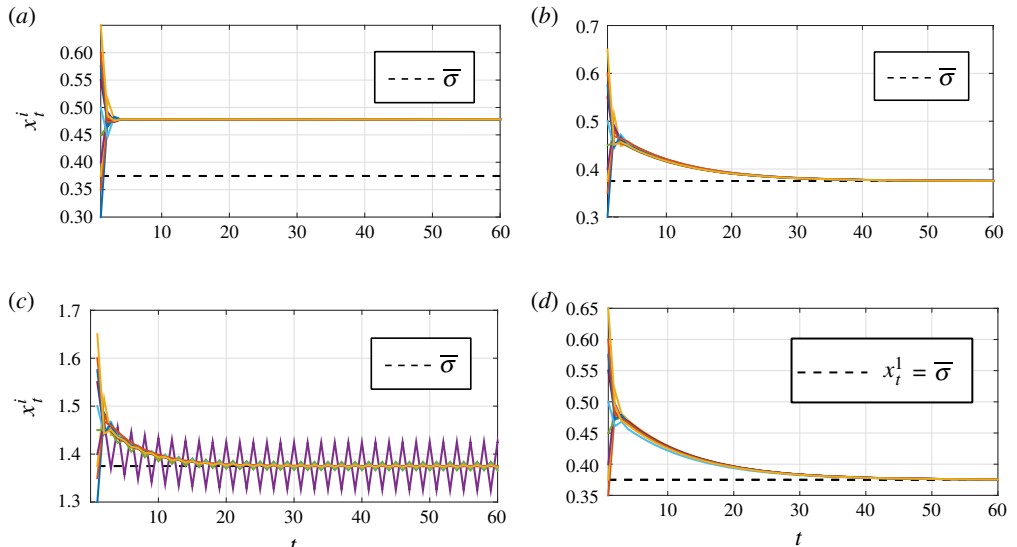

**Figure 2.** Evolution of the agents' dynamics (4.2): (*a*) without learning, (*b*) with learning and $\epsilon_i$ satisfying the conditions of theorem 4.1, (*c*) with learning and $\epsilon_i$ not satisfying the conditions of theorem 4.1, and (*d*) evolution of the agents' dynamics with a leader (4.5).

**Theorem 5.3.** *Consider the opinion dynamics in* (5.4) *and assume that the matrix $\tilde{A}$ is substochastic and irreducible. Then consensus to $\mathbf{1}_n \otimes \bar{\sigma}$ is reached, i.e.* $\lim_{t \to \infty} X_t = \mathbf{1}_n \otimes \bar{\sigma}$.

The proof of theorem 5.3 follows the same line of reasoning as the proof of theorem 4.4 and it is omitted here.

**Corollary 5.4.** *Let $\| \cdot \|_*$ denote some matrix norm such that $\|\tilde{A}\|_* < 1$. Then consensus to $\bar{\sigma}$ is reached exponentially with convergence rate $\|\tilde{A} \otimes I_k\|_*$, i.e.* $\|X_t - (\mathbf{1}_n \otimes \bar{\sigma})\|_\infty \leq C\|\tilde{A} \otimes I_k\|_*^{t-1}\|X_1 - (\mathbf{1}_n \otimes \bar{\sigma})\|_\infty$, *for some positive constant $C \in \mathbb{R}_{>0}$.*

# 6. Numerical simulations

Consider the opinion dynamics with feedback (4.2) with 10 agents ($n = 10$), $\bar{\sigma} = 0.375$ and initial condition

$$X_1 = (0.3, 0.35, 0.37, 0.4, 0.45, 0.5, 0.55, 0.57, 0.6, 0.65)^\top.$$

In both exchange-based and OTC markets, it is easy to ascertain who the main market-makers are for options on single stock or commodity [79,80]. Option market-makers are usually investment banks and big trading houses. In this sense, the number of players is not large and thus the models developed always have a finite number of agents, $N = 10$.

Figure 2 depicts the obtained simulation results for different values of the learning parameters $\epsilon_i$, $i = 1, \ldots, 10$. Specifically, figure 2*a* shows results without learning, i.e, $\epsilon_i = 0$ (here there is no consensus to $\bar{\sigma}$), figure 2*b* depicts the results for $\epsilon_i = 0.9a_{ii}$. As stated in theorem 4.1, consensus to $\bar{\sigma}$ is reached. Figure 2*c* shows results for $\epsilon_i = 0.9a_{ii} + 0.94\,b_i$ with $b_4 = 1$ and $b_i = 0$ otherwise, $i = 1, \ldots, 10$. Note that, in this case, the value of $\epsilon_4$ violates the condition of theorem 4.1 (i.e. $\epsilon_4 \notin (0, a_{44})$) and, as expected, consensus is not reached. Next, consider the opinion dynamics with a leader (4.5) with $n = 10$ and initial condition

$$X_1 = (\bar{\sigma}, 0.35, 0.37, 0.4, 0.45, 0.5, 0.55, 0.57, 0.6, 0.65)^\top.$$

For the leader case, the opinion weights matrix is constructed by replacing the first row of $A$ by $(1, 0, \ldots, 0)$. The corresponding matrix $\tilde{A}$ (defined in 4.5) is substochastic and irreducible, and $\sum_{i=2}^{i=10} a_{ij} < 1$, $j = 1, \ldots, 10$. Hence, all the conditions of theorem 4.4 are satisfied and consensus to $\bar{\sigma} = 0.375$ is reached. Figure 2*d* shows the corresponding simulation results. Finally, figure 3 shows the evolution of the vectored opinion dynamics (5.2) with $n = 10$ and $k = 3$ (i.e. 10 three-dimensional agents), matrix $A$ as in the case with feedback, (vectored) volatility $\bar{\sigma} = (0.67, 0.22, 0.88)^\top$, learning parameters $\epsilon_i = 0.9a_{ii}$ for $a_{ii}$ as in $A$, and initial condition $\mathbf{1}_k \otimes X_1$ with $X_1$ as in the first experiment above.

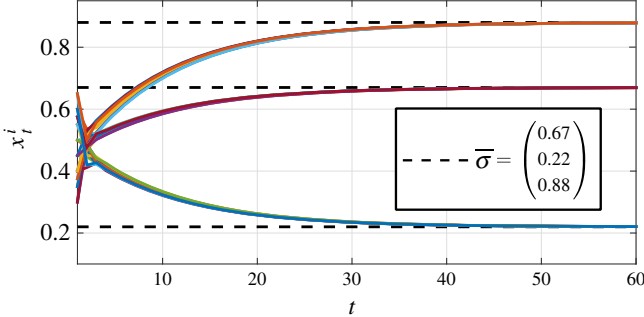

**Figure 3.** Evolution of the multi-dimensional agents' dynamics with learning (5.2).

# 7. Arbitrage bounds

We have taken the true volatility parameter as exogenous to our models. Our only requirement is that there is no static arbitrage, by which we mean that all the quotes in volatility which translate to option prices are such that one cannot trade in the different strikes to create a profit. Checking whether a volatility surface is indeed arbitrage-free is non-trivial, nevertheless some sufficient conditions are well known [81–83]. As long as the volatility surface satisfies them our analysis implies global stability towards an arbitrage-free smile.

We parametrize the volatility function (assuming expiry $T$ and $S_0$ are fixed) and denote the option price as

$$\overline{BS}(K, \sigma(K)) \triangleq BS(S_0, K, T, \sigma(K)).$$

Our attention is on varying $K$, to ensure no static arbitrage. We assume that the $\sigma(K)$ translates into unique call option dollar prices. This follows from the strictly positive first derivative of the option price formula with respect to $\sigma$. We require two conditions:

— **Condition 1: (Call Spread)** For $0 < K_1 \leq K_2$, we have $\overline{BS}(K_1, \sigma(K_1)) \geq \overline{BS}(K_2, \sigma(K_2))$.
— **Condition 2: (Butterfly Spread)** For $0 < K_1 < K_2 < K_3$, $\overline{BS}(K_1, \sigma(K_1)) + ((K_2 - K1)/(K_3 - K_2)) \times \overline{BS}(K_3, \sigma(K_3)) \geq (K_3 - K1)/(K_3 - K_2) \times \overline{BS}(K_2, \sigma(K_2))$.

Assa *et al.* [84] examine the case of checking static arbitrage conditions, using machine learning techniques; moreover, their notion of quotes being arbitrage-free is extended to exclude calendar spread arbitrage across different maturities. We highlight the conditions needed for a single slice of the volatility surface as $T$ is fixed in our environment. How arbitrage-free curve volatility conditions are developed is not an easy task: see the extended accounts in [32,84–88].

# 8. Discussion

## 8.1. Future work

Social learning is an active area of research in many different fields. By combining aspects of social learning models with dynamical systems, we were able to develop insightful analysis for the volatility smile. This can be extended further. There are several immediate possibilities. Can the number of strikes be infinite? We restricted the models to a finite number of strikes: fixed $k$. In practical terms, at any given time, there are usually two strikes below and two strikes above the ATM level that are liquid. This means the corresponding quotes are visible or updated for five strikes. One way to circumvent this is to consider arbitrage-free volatility curves. But again, we are faced with the observational nature of our framework. A trader only observes a fixed number of strikes of his competitors. The issue of how to introduce heterogeneity in the volatility curves, which themselves emanate from specific pricing models, remains open.

The number of agents can also be infinite. Perhaps a propagation of chaos type of result could shed some light on how an individual trader interacts with the mean-field limit [89–91]. In this case, we lose the heterogeneity of beliefs and the behaviour we are trying to study would have a different implication.

Moreover, considerable technical machinery is required [92,93]. We could study the pure limiting behaviour as $t, n \to \infty$. In our current framework, this would have to be balanced with whether an individual can observe an infinite number of competitors. While the technical subtleties are not insurmountable, the modelling issues are more subjective.

The technical issues in random matrix products, briefly discussed in this paper, assure us that much more work needs to be done on the modelling and mathematical front. For example, the matrices $A$ and $\mathcal{E}$ can be dependent with correlation decreasing in time. Work in this direction has been addressed by Popescu & Vaidya [94].

## 8.2. Connection

Recently, there has been some rather interesting work at the intersection of computer science and option pricing. Demarzo *et al.* [95] showed how to use efficient online trading algorithms to price the current value of financial instruments, deriving both upper and lower bounds using online trading algorithms. Moreover, Abernethy *et al.* [96,97] developed a BS price as sequential two-player zero-sum game. While these papers made an excellent start to bridge the gap between two different academic communities—mainly mathematical finance and theoretical computer science—they do not address the reality of volatility smiles and trading. Our contribution can be viewed as making these connections more concrete. The smile itself is a conundrum and there have even been articles questioning whether it can be solved [98]. The traditional way from the ground up is to develop a stochastic process for the volatility and asset price, possibly introducing jumps or more diffusions through uncertainty [99,100]. Such models have been successfully developed, but the time is ripe to incorporate multi-agent models with arbitrage-free curves.

Introducing learning agents in stochastic differential equation models [101], such as the BS model, is an exciting proposition. Moreover, opinion dynamics as a subject on its own has been studied quite extensively. Recent references that present an expansive discussion in computer science are [8,102]. Econophysics is the right community to develop new models. After all, there is no attachment to utilities of players or stochastic volatility models so entrenched in the mathematical finance community. Free from these shackles, researchers can use a range of tools and techniques to build more sophisticated models. Moreover, there is no restriction or debate on continuous or discrete time. While our framework is discrete, continuous time could perhaps show a way forward to incorporate models from mathematical finance and financial economics [103–105]. Jarrow [106] makes the case for continuous time, arguing that today's financial markets trade and update at high frequency.

In this paper, we introduce models of learning agents in the context of option trading. A key open question in this setting is how the market comes to a consensus about market volatility, which is reflected in derivative pricing through the BS formula. The framework we have established allows us to explore other areas. Thus far, we took the smile as an exogenous object, proving convergence to equilibrium beliefs. A natural step forward would be to look at the beliefs as probability measures, where each measure corresponds to a different option pricing model. Our learning models focus on interaction between agents. Actually, agents can be interpreted as algorithms. Each algorithm corresponds to a particular belief of a pricing model. Until now, the replication paradigm has led to very sophisticated models. The future may belong to deep hedging arguments [107]. Still, whether we consider models or algorithms, interaction will always be a topic of interest.

Data accessibility. This article has no additional data. Code for simulations is available within the Dryad Digital Repository: https://doi.org/10.5061/dryad.prr4xgxjg [108].
Authors' contributions. T.V. conceptualized the model. T.V. and C.M. formalized the mathematical framework. G.P. guided the work and aided the discussions and structuring of the manuscript. T.V. and C.M. wrote the manuscript.
Competing interests. We declare we have no competing interests.
Funding. T.V. acknowledges a SUTD Presidential fellowship. C.M. acknowledges the National Research Foundation (NRF), Prime Minister's Office, Singapore, under its National Cybersecurity R&D Programme (Award no. NRF2014NCR-NCR001-40) and administered by the National Cybersecurity R&D Directorate. G.P. acknowledges AcRF Tier 2 grant nos. 2016-T2-1-170, PIE-SGP-AI-2020-01, NRF2019-NRF-ANR095 ALIAS grant and NRF2018 Fellowship NRF-NRFF2018-07.
Acknowledgements. The authors thank Ioannis Panageas, Ionel Popescu, Niels Nygaard and JM Schumacher for fruitful discussions.

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
