## [Reviewer comments · Royal Society Open Science]

Review History

RSOS-201188.R0 (Original submission)

Review form: Reviewer 1

Is the manuscript scientifically sound in its present form?

Yes

Are the interpretations and conclusions justified by the results?

Yes

Is the language acceptable?

Yes

Do you have any ethical concerns with this paper?

No

Have you any concerns about statistical analyses in this paper?

No

Recommendation?

Accept with minor revision (please list in comments)

Comments to the Author(s)

In the paper entitled "Learning Agents in Black-Scholes Financial Markets," the authors have proposed a few agent-based models, in which the agents update their beliefs about the true

implied volatility associated with the Black-Scholes model, based on the opinions of other traders. The paper critically reviews the merits and demerits of the current Econophysics literature that considers the volatility in the Black-Scholes model to be constant across all strikes. There are two scenarios considered: one with consensus (scalar agent dynamics) and the other with consensus (vector agent dynamics). They further consider cases, where they have: (a) feedback, or (b) random, or (c) consensus with an unknown leader. In one model, the difference between the agent's opinion and volatility is incorporated in a naive strategy through a learning coefficient. In another model, information about external opinion is not included; instead it incorporates information of an unknown leader. The authors perform convergence analyses for the proposed opinion dynamics and derive the convergence results under different conditions. These results are further generalized later. Numerical experiments are performed to validate the models, which exhibit that convergence is guaranteed only when the opinion dynamics fulfills certain hypotheses as derived theoretically. The topic is certainly interesting to explore. The opinion dynamics proposed by the authors are non-trivial yet simple to implement; further, they provide convergence analysis to guarantee consensus to the implied volatility. The methodology is sound and the results are very interesting. I have a few comments / suggestions: (i) The paper can be written in a more organized and clear manner, as sometimes it gets confusing. (ii) There are several misspellings, typographical and grammatical errors in the paper, e.g., name of author (Chakroborti -> Chakraborti) and repetition of words ('the', appears twice at places). (iii) At times, informal language has been used (e.g., don't). (iv) The last section should be expanded to include some critical discussions. Hence, I recommend the acceptance of the paper with minor revision.

Review form: Reviewer 2

Is the manuscript scientifically sound in its present form?

Yes

Are the interpretations and conclusions justified by the results?

Yes

Is the language acceptable?

Yes

Do you have any ethical concerns with this paper?

No

Have you any concerns about statistical analyses in this paper?

No

Recommendation?

Major revision is needed (please make suggestions in comments)

Comments to the Author(s)

The paper is well written. But it lacks any argument to make a comparison with other literature. In particular, there are a few papers to model the volatility smile. For instance, the following work uses Machine Learning methods to learn implied volatility:

H Assa, M Pouralizadeh, A Badamchizadeh, (2019) Sound deposit insurance pricing using a machine learning approach, Risks 7 (2), 45

On the other hand, I think there is confusion between market incompleteness and imperfection. Markets can be incomplete even if there is no uncertainty. Actually, incompleteness is due to a

lack of enough instruments to perfectly replicate the market assets. This means you may know the market model (probability) but have no enough assets to hedge. While uncertainty is not about that. Examples of incompleteness include the jump-diffusion price process, while for uncertainty one can consider an interval for the volatility values (due to confidence interval of estimation).

Decision letter (RSOS-201188.R0)

Dear Dr Vaidya

On behalf of the Editors, we are pleased to inform you that your Manuscript RSOS-201188 "Learning Agents in Black-Scholes Financial Markets" has been accepted for publication in Royal Society Open Science subject to minor revision in accordance with the referees' reports. Please find the referees' comments along with any feedback from the Editors below my signature.

Please submit your revised manuscript and required files (see below) no later than 7 days from today's (ie 03-Sep-2020) date. Note: the ScholarOne system will 'lock' if submission of the revision is attempted 7 or more days after the deadline. If you do not think you will be able to meet this deadline please contact the editorial office immediately.

on behalf of Dr Robert MacKay (Associate Editor) and Mark Chaplain (Subject Editor)
openscience@royalsociety.org

Associate Editor Comments to Author (Dr Robert MacKay):

Associate Editor: 1

Comments to the Author:

The reviewers both found the paper interesting. One recommended major revision. Actually I think their suggestions are relatively minor but they should indeed be addressed. The other

recommended to accept with minor revision. Their suggestions are indeed ones that should be addressed. Based on these, my recommendation is to accept subject to minor revisions.

Reviewer comments to Author:

Reviewer: 1

Comments to the Author(s)

In the paper entitled "Learning Agents in Black-Scholes Financial Markets," the authors have proposed a few agent-based models, in which the agents update their beliefs about the true implied volatility associated with the Black-Scholes model, based on the opinions of other traders. The paper critically reviews the merits and demerits of the current Econophysics literature that considers the volatility in the Black-Scholes model to be constant across all strikes. There are two scenarios considered: one with consensus (scalar agent dynamics) and the other with consensus (vector agent dynamics). They further consider cases, where they have: (a) feedback, or (b) random, or (c) consensus with an unknown leader. In one model, the difference between the agent's opinion and volatility is incorporated in a naive strategy through a learning coefficient. In another model, information about external opinion is not included; instead it incorporates information of an unknown leader. The authors perform convergence analyses for the proposed opinion dynamics and derive the convergence results under different conditions. These results are further generalized later. Numerical experiments are performed to validate the models, which exhibit that convergence is guaranteed only when the opinion dynamics fulfills certain hypotheses as derived theoretically. The topic is certainly interesting to explore. The opinion dynamics proposed by the authors are non-trivial yet simple to implement; further, they provide convergence analysis to guarantee consensus to the implied volatility. The methodology is sound and the results are very interesting. I have a few comments /suggestions: (i) The paper can be written in a more organized and clear manner, as sometimes it gets confusing. (ii) There are several misspellings, typographical and grammatical errors in the paper, e.g., name of author (Chakroborti -> Chakraborti) and repetition of words ('the', appears twice at places). (iii) At times, informal language has been used (e.g., don't). (iv) The last section should be expanded to include some critical discussions. Hence, I recommend the acceptance of the paper with minor revision.

Reviewer: 2

Comments to the Author(s)

The paper is well written. But it lacks any argument to make a comparison with other literature. In particular, there are a few papers to model the volatility smile. For instance, the following work uses Machine Learning methods to learn implied volatility:

H Assa, M Pouralizadeh, A Badamchizadeh, (2019) Sound deposit insurance pricing using a machine learning approach, *Risks* 7 (2), 45

On the other hand, I think there is confusion between market incompleteness and imperfection. Markets can be incomplete even if there is no uncertainty. Actually, incompleteness is due to a lack of enough instruments to perfectly replicate the market assets. This means you may know the market model (probability) but have no enough assets to hedge. While uncertainty is not about that. Examples of incompleteness include the jump-diffusion price process, while for uncertainty one can consider an interval for the volatility values (due to confidence interval of estimation).

===PREPARING YOUR MANUSCRIPT===

Your revised paper should include the changes requested by the referees and Editors of your manuscript. You should provide two versions of this manuscript and both versions must be provided in an editable format: one version identifying all the changes that have been made (for instance, in coloured highlight, in bold text, or tracked changes);a 'clean'

version of the new manuscript that incorporates the changes made, but does not highlight them. This version will be used for typesetting.

===PREPARING YOUR REVISION IN SCHOLARONE===

Author's Response to Decision Letter for (RSOS-201188.R0)

See Appendix A.

Decision letter (RSOS-201188.R1)

Dear Dr Vaidya,

It is a pleasure to accept your manuscript entitled "Learning Agents in Black-Scholes Financial Markets" in its current form for publication in Royal Society Open Science.

Due to rapid publication and an extremely tight schedule, if comments are not received, your paper may experience a delay in publication. Royal Society Open Science operates under a continuous publication model. Your article will be published straight into the next open issue and this will be the final version of the paper. As such, it can be cited immediately by other researchers. As the issue version of your paper will be the only version to be published I would

advise you to check your proofs thoroughly as changes cannot be made once the paper is published.

on behalf of Dr Robert MacKay (Associate Editor) and Mark Chaplain (Subject Editor)
openscience@royalsociety.org
